# Protective Effect of Inactivated COVID-19 Vaccines against Progression of SARS-CoV-2 Omicron and Delta Variant Infections to Pneumonia in Beijing, China, in 2022

**DOI:** 10.3390/vaccines10081215

**Published:** 2022-07-29

**Authors:** Juan Li, Rui Song, Zheng Yuan, Zheng Xu, Luodan Suo, Qing Wang, Yuan Li, Yanlin Gao, Xiaomei Li, Xiaoyou Chen, Jiang Wu

**Affiliations:** 1Beijing Center for Disease Prevention and Control, Beijing Research Center for Preventive Medicine, Beijing 100013, China; taozilijuan@126.com (J.L.); suoluodan2004@163.com (L.S.); liyuan@bjcdc.org (Y.L.); gaoyl@bjcdc.org (Y.G.); anananlxm@163.com (X.L.); 2Beijing Ditan Hospital, Capital Medical University, Beijing 100013, China; songruii@hotmail.com (R.S.); ben_nk@139.com (Z.Y.); 3Beijing Municipal Health Commission, Beijing 100101, China; xuzheng@wjw.beijing.gov.cn; 4Xicheng Distract Center for Diseases Control and Prevention, Beijing 100120, China; wwq97@163.com

**Keywords:** COVID-19, Omicron, Delta, pneumonia, inactive vaccine, effect

## Abstract

This real-world study explores the effect of coronavirus disease 2019 (COVID-19) inactivated vaccines on the prevention of asymptomatic or mild Delta or Omicron variant infections progressing to pneumonia. Association between severe acute respiratory syndrome coronavirus 2 (SARS-CoV-2) pneumonia cases and vaccination was measured with a multivariable logistic regression, stratified by genotype and age groups. We recruited 265 cases (111 (41.9%) infected with Delta and 154 (58.1%) with Omicron variants). There were 22 asymptomatic infected individuals, 156 mild cases without pneumonia, and 87 moderate cases with pneumonia. There was a markedly increased risk of progression to pneumonia in Delta infected cases, unvaccinated, or partially vaccinated COVID-19 patients with diabetes and those aged ≥60 years. Patients who had completed booster doses of inactivated vaccines had a reduced risk of 81.6% (95% CI: 55.6–92.4%) in progressing to pneumonia over those who were unvaccinated or partially vaccinated. The risk of progressing to pneumonia was less reduced by 88.7% (95% CI: 56.6–97%) and 73.9% (95% CI: 1.4–93.1%) among Delta and Omicron-infected patients, and was reduced by 78.5% (95% CI: 45.3–91.6%) and 94.1% (95% CI: 21.5–99.6%) among patients aged <60 and ≥60 years, respectively. Our data indicated that a complete vaccination with a booster reduced the risk of asymptomatic or mild Delta or Omicron variant COVID-19 progressing to pneumonia and, thus, reduced the pressure of severe illness on medical resources.

## 1. Introduction

The implementation of coronavirus disease 2019 (COVID-19) vaccines against severe acute respiratory syndrome coronavirus 2 (SARS-CoV-2) is a major asset in slowing down the COVID-19 pandemic [1]. At the time of writing, more than 250 vaccines have been developed and 67 vaccines have been evaluated in phase III clinical trials, according to the World Health Organization (WHO) [2]. Five types of vaccine development techniques have been established to produce COVID-19 vaccines in China: inactivated whole-virus vaccines, adenovirus vectored vaccines, recombinant protein subunit vaccines, nucleic acid vaccines, and attenuated influenza virus vectored vaccines [3]. Since December 2020, a massive vaccination campaign has been underway worldwide. China initiated a mass COVID-19 vaccination campaign on 15 December 2020. By 14 July 2022, the cumulative number of vaccinated doses exceeded 3.41 billion, with over 90% coverage [4]. Inactivated vaccines are the most widely used vaccines in China. The COVID-19 epidemic is still complicated and challenging in China with localized cases and outbreaks still occurring [5]. Knowledge of the real-world protective effect of COVID-19 vaccines is critically important for shaping strategy and policy for future programs [6]. The increasingly accumulated evidence suggests that inactivated COVID-19 vaccines could protect from SARS-CoV-2 infection, although not wholly, and prevent severe illness progression efficiently [7,8]. However, as SARS-CoV-2 variants are characterized by their transmissibility, disease severity, and ability to evade humoral immunity [9,10,11], every opportunity of evaluation on vaccine effectiveness in the real world is important to guide policy decisions.

Following the confirmed COVID-19 cases imported from endemic areas outside its borders, Beijing reported COVID-19 cases one after another from January 2022. Delta and Omicron variants caused two distinct transmission chains from January to April of 2022 in Beijing. The local cases infected by the Omicron variant were the first to be detected and transmitted in Beijing, which was first identified in South Africa, in November, 2020, and now has spread globally [12]. Assessing COVID-19 vaccine performance against the rapidly spreading SARS-CoV-2 Delta and Omicron variants is critical to inform public health guidance. Compared with the original type and Delta variant of SARS-CoV-2, the Omicron variant has an apparent increased transmission and immune escape ability [13]. Regarding observational real-life data, primary immunization with COVID-19 vaccines seems to effectively prevent SARS-CoV-2 infection against the original strain, Alpha, and Beta variants, but with reduced effectiveness against the Delta strain in many studies. A rapid increase in COVID-19 cases due to the Omicron variant of SARS-CoV-2 in vaccinated populations has led to concerns about the effectiveness of current vaccines. It is worth considering whether the widely used inactivated vaccines are still effective for Delta and Omicron variants [14]. Thus, we performed a real-world study using patients’ clinical and epidemiological data during the first four months of 2022. The aim of this study was to determine to what degree the inactivated vaccines could prevent asymptomatic and mild cases of SARS-CoV-2 Delta and Omicron variant infections progressing to pneumonia.

## 2. Materials and Methods

### 2.1. Study Design and Participants

We recruited 265 asymptomatic infected persons and patients infected with COVID-19 Delta and Omicron variants from January to April of 2022 in Beijing. We collected the data of each patient, including demographic characteristics, medical history, vaccination status, chronic underlying diseases, treatments, outcomes, and virus genotype. The onset data were defined as the first appearance of symptoms in patients or the first positive detection of the virus with SARS-CoV-2 nucleic acid test in asymptomatic infected persons.

### 2.2. Vaccination Status

Vaccinations were considered valid only if they had been documented in the national or provincial Immunization Information System. Participants were categorized into an unvaccinated group, a partially vaccinated group, a completed primary vaccination group, and a completed booster vaccination group based on compliance with China CDC’s COVID-19 Vaccines Immunization Schedule. The unvaccinated group consisted of individuals who did not receive any COVID-19 vaccines before their last known contact with a confirmed case. The partially vaccinated group consisted of individuals who had received either one dose of COVID-19 inactivated vaccines or had received two doses of inactivated vaccines with receipt of the second dose less than 14 days before exposure to an infected individual. The completed primary vaccination group consisted of individuals who completed two doses of inactivated vaccines 14 days or more before exposure to an infected individual. The completed booster vaccination group consisted of individuals who completed the third dose of inactivated vaccines 7 days or more before exposure to an infected individual.

### 2.3. Outcomes

We evaluated three outcomes: asymptomatic COVID-19, mild COVID-19, and moderate COVID-19. Case classifications were based on China’s COVID-19 Prevention and Control Protocol (ninth edition) [15] and COVID-19 Diagnosis and Treatment Protocol (Trial ninth edition) [16]: asymptomatic, mild, moderate, severe, and critically severe. There were no clinical manifestations in asymptomatic patients, only the nucleic acid test for SARS-CoV-2 virus was positive. Mild cases with mild clinical symptoms were found to have no pneumonia on imaging. Pneumonia symptoms included both clinical and imaging manifestations in moderate cases. Clinical manifestations and indicators were more serious in severe and critical cases than in moderate cases. COVID-19 pneumonia in this study included moderate, severe, and critically severe cases with evidence of pneumonia. All confirmed cases, including mild cases, were admitted to designated medical institutions, and all asymptomatic infected individuals were admitted to the square cabin hospitals.

### 2.4. Statistical Analysis 

Continuous variables were expressed as the mean (95% confidence interval (CI)) and categorize variables as frequency. As appropriate, Chi-square, Mann–Whitney U, or Fisher’s exact tests were used to compare groups. The relationship between COVID-19 pneumonia and factors was analyzed using univariate and multivariate regression analysis, and odds ratios and 95% confidence intervals were calculated. Genotype- and age-stratified subgroup analysis was also performed. The significance level was set at 0.05. All analyses were performed using SPSS software.

## 3. Results

### 3.1. Basic Characteristics

In total, 265 hospitalized patients were included in the analysis, of which 22 (8.3%) were asymptomatic, 156 (58.9%) were mild cases without pneumonia, and 87 (32.8%) were moderate cases with pneumonia. Over 55.5% (147/265) of patients were male. The mean age of patients was 40.7 (95% CI: 38.6–42.8). Genome sequencing confirmed that 111 (41.9%) were infected with the Delta variant and 154 (58.1%) were infected with the Omicron variant. Further, 227 patients were vaccinated with inactivated vaccines, of which 154 received CoronaVac (Sinovac’s inactivated vaccine) and 73 received BBIBP-CorV (Sinopharm’s inactivated vaccine) (Table 1).

### 3.2. Risk Factors of Patient Progression to Pneumonia

The 265 infected individuals were categorized as COVID-19 pneumonia cases and non-pneumonia cases. The proportion of Delta-infected patients was significantly higher in patients with pneumonia (71.3%) than in patients without pneumonia (27.5%). The proportion of Omicron-infected patients was significantly lower in patients with pneumonia (28.7%) than in patients without pneumonia (72.5%) (χ^2^ = 45.922, *p* < 0.001). The relative risk of progression to pneumonia for Delta-infected patients was 13 (95% CI: 6.263–26.982) times more than for Omicron-infected patients. The proportion of elderly people aged ≥60 years was higher in patients with pneumonia (18.4%) than in patients without pneumonia (6.7%) (χ^2^ = 8.382, *p* = 0.004). The relative risk of progression to pneumonia among patients aged ≥60 years was 5.902(95% CI: 1.852–18.809) times more than among patients aged <60 years. In patients with pneumonia, there was a higher proportion of diabetes patients (16.1%) than in patients without pneumonia (3.4%) (χ^2^ = 13.552, *p* < 0.001). The relative risk of progression to pneumonia for diabetes patients was 4.888 (95% CI: 1.417–16.856) times more than for non-diabetes patients. It was also found that there was a higher proportion of hypertension patients in patients with pneumonia (21.8%) than in patients without pneumonia (10.7%) (χ^2^ = 5.930, *p* = 0.015) (Table 2).

### 3.3. Protective Effect of Inactivated COVID-19 Vaccines

After adjusting for the potential confounders, including genotype, age, diabetes, and hypertension, compared with the unvaccinated or partially vaccinated group, those who had completed the booster vaccination had a significantly decreased risk of pneumonia by 81.6% (OR: 0.184, 95% CI: 0.076–0.444, *p* < 0.001; risk reduction: 81.6%, 95% CI: 55.6%–92.4%). The risk of progression to pneumonia was also decreased for the patients who had complete primary vaccination, but not significantly (OR: 0.650, 95% CI: 0.257 to 1.643) (Figure 1).

By genotype grouping, compared with the unvaccinated or partially vaccinated group, the risk of progression to pneumonia was reduced by 88.7% (OR: 0.113, 95% CI: 0.030–0.434, *p* = 0.001; risk reduction: 88.7%, 95% CI: 56.6–97.0%) in patients with the Delta variant and was also decreased by 73.9% (OR: 0.261, 95% CI: 0.069–0.986, *p* = 0.048; risk reduction: 73.9%, 95% CI: 1.4–93.1%) in patients with the Omicron variant who had completed the booster vaccination.

By age grouping, compared with the unvaccinated or partially vaccinated group, the risk of progression to pneumonia was reduced by 78.5% (OR: 0.215, 95% CI: 0.084–0.547, *p* = 0.001; risk reduction: 78.5%, 95% CI: 45.3%–91.6%) in patients aged <60 years and was also decreased by 94.1% (OR: 0.059, 95% CI: 0.004–0.785, *p* = 0.032; risk reduction: 94.1%, 95% CI: 21.5%–99.6%) in patients aged ≥60 years who had completed the booster vaccination.

In addition, the onset time since last vaccine dose was significantly different between the group that received primary vaccination and the group that received booster vaccination. The interval time in complete primary vaccinated cases (188.6 days, 95% CI: 168.8–208.5) was longer than that in booster vaccinated cases (92.0 days, 95% CI: 85.9–98.1).

### 3.4. Effect of Vaccination on Medical Resources

The duration of hospitalization in patients with pneumonia (17.9 days, 95% CI: 16.6–19.2) was longer than that in patients without pneumonia (15.3 days, 95% CI: 14.6–16.0) (*p* = 0.001). The proportion of patients receiving intravenous drugs was higher in the COVID-19 pneumonia group than in the no-pneumonia group (35.6% vs. 2.8%, *p* < 0.001). The proportion of patients receiving oxygen treatment in the COVID-19 pneumonia group was higher than in the no-pneumonia group (9.2% vs. 1.1%, *p* = 0.004) (Table 1). The difference in duration of hospitalization among people with different vaccination status was statistically significant for pneumonia patients (*p* = 0.009). Unvaccinated patients with pneumonia (21.7 days, 95% CI: 18.7–24.7) stayed in the hospital longer than pneumonia patients with complete primary (17.7 days, 95% CI: 15.4–20.1) and booster vaccination (16.1 days, 95% CI: 14.3–18.0) (*p* = 0.022 between unvaccinated and primary vaccinated patients, *p* = 0.001 between unvaccinated and booster vaccinated patients). Additionally, in pneumonia patients with different vaccination status, the proportion of patients treated intravenously was significantly different (*p* = 0.012). In unvaccinated pneumonia patients, the proportion of patients treated intravenously was higher than in pneumonia patients with complete primary and booster vaccination (Table 3).

## 4. Discussion

The Omicron variant of SARS-CoV-2 is more transmissible than other strains and is now the primary cause of the COVID-19 outbreak [17,18]. In December, 2021, the first instance of the Omicron variant occurred in Tianjin, China. The first report of a local confirmed Omicron variant case was in Beijing in January, 2022. It is possible that variants of Omicron are capable of evading vaccine immunity resulting from vaccination and previous infections. Neutralizing antibodies’ potency is blunted more extensively with the Omicron variant than other variants, which reduces the effectiveness of existing vaccines. Despite massive vaccinations, it is becoming increasingly common for the Omicron variant to cause infections [19]. Hospitalizations and serious illnesses can still be reduced by COVID-19 vaccines, despite their reduced effectiveness with the Omicron variant [20]. At the beginning of the rapid spread of the Omicron variant and during co-transmission with the Delta variant in Beijing, assessing COVID-19 vaccine performance on preventing progression from asymptomatic or mild cases to pneumonia is critical to inform public health guidance.

Infections with Omicron are reported to be less severe than infections with previous variants, such as Delta [21,22]. Compared with the 3075 Delta cases reported in France, significantly fewer patients were hospitalized, needed intensive care, and died in the first 1119 Omicron cases [23]. Similar conclusions could be drawn from the present research, which showed the risk of progression to pneumonia was 6–27-times higher in those infected with the Delta variant than in those infected with the Omicron variant.

Comorbidities and old age are known risk factors associated with severe illness in SARS-CoV-2 patients [24,25]. An increased risk of death was associated with aging, cardiovascular disease, diabetes, chronic respiratory disease, hypertension, and cancer [26]. A meta-analysis of 76 studies, including 17,860,001 patients across 14 countries, indicated that those with the most severe disease were more likely to have hypertension, respiratory disease, and cardiovascular disease [27]. Our study, though with a limited number of cases, was consistent with previous findings. Patients aged ≥60 years and with diabetes had an increased risk of progression from asymptomatic or mild COVID-19 infection to pneumonia.

As of now, existing COVID-19 vaccines still provide a reasonable level of protection. The boosters are expected to be effective in strengthening immunity against Delta and Omicron variants. It is likely that the booster dose will prevent antibodies from evading, since there will be an increase in neutralizing antibodies [28,29]. According to recent studies, the Omicron variant is substantially less effective against symptomatic disease compared to Delta variant. The effectiveness of vaccines declined rapidly after the second dose, while booster doses significantly increased protection against mild infections. There was little evidence that Omicron provided protection against severe disease, due to the small number of hospitalizations associated with Omicron infection. This study, involving 265 patients in the hospital, showed that the risk of progression from asymptomatic or mild cases to pneumonia cases had substantially decreased in completely booster vaccinated patients. After adjusting for genotype, age, and underlying medical conditions, the risk reductions remained significant at 81.6%, especially with apparent risk reduction of 94.1% in patients aged ≥60 years. Our findings suggest that three doses of COVID-19 inactivated vaccines conferred protection over being unvaccinated and receiving one dose against progression to pneumonia, caused by both the Delta and Omicron variants. These findings provide evidence supporting the fact that primary doses alone are less protective than booster doses against progression to pneumonia, since the interval time since last vaccine dose in complete primary vaccinated cases was longer than that in booster vaccinated cases. The timing effect may be a reason that two doses of vaccination were insufficient to prevent pneumonia progression. In addition, Omicron has a higher OR than Delta, suggesting booster doses are less protective, although it is not statistically significant, which is in agreement with in vitro neutralization studies suggesting Omicron may be capable of immune evasion [30,31]. A higher booster coverage rate may be required for achieving the same public health benefits as during Delta dominance in the wave of Omicron.

Emerging laboratory data indicate there is a substantial reduction in neutralizing antibody response to Delta and Omicron variants compared with the original strains, which may explain the reduced vaccine effectiveness in vaccinated persons [32]. COVID-19 vaccines have been more effective against severe disease, including hospitalization and death, and cellular immune responses are likely to play a relatively greater role in protection [33,34].

Prevention of severe disease is the most important goal of vaccination, in order to avoid a serious drain on medical and health resources. After the outbreak of COVID-19 in many areas, the medical resources were severely limited, especially the number of hospital beds, often being insufficient. It is possible to reduce the demand for medical resources, including the availability of beds, by reducing hospital stays [35]. This study showed that pneumonia cases required more medical resources, such as longer hospitalization, and a higher proportion of patients requiring oxygen therapy and intravenous medication. The reason for longer hospitalization in pneumonia cases was due to the discharge standard to a certain extent. The discharge criteria for the confirmed cases included more than just negative nucleic acid tests of two consecutive samples, also including a significant improvement in clinical manifestations, such as the temperature returning to normal for over 3 days, and respiratory symptoms and pulmonary imaging improving significantly. This study also showed that unvaccinated pneumonia cases stayed in the hospital longer than pneumonia cases with complete primary or booster vaccination. In addition, unvaccinated pneumonia patients received more intravenous treatment than pneumonia patients who had received primary or booster vaccinations. Therefore, vaccination can significantly reduce the risk of developing pneumonia after infection, thus, reducing the drain on medical resources.

This study has some limitations. Firstly, due to the data that all participants had confirmed COVID-19, we were unable to estimate vaccine effectiveness against Delta and Omicron variant infections. Secondly, due to the observational nature of this study, it is possible that there were unmeasured confounding variables that affected the results. Thirdly, the small number of cases makes it impossible to decompose estimates based on population characteristics that may influence vaccine effectiveness. Finally, rigorous non-pharmaceutical interventions may have affected the results in unknown directions.

## 5. Conclusions

Our findings indicated that completing two doses of vaccination with inactivated COVID-19 vaccines is insufficient to adequately protect against Delta and Omicron variant infection progressing to pneumonia. A booster dose significantly increased protection against progressing to pneumonia. Boosters are likely to provide even greater protection against severe and fatal diseases. Our findings support that third doses of COVID-19 inactivated vaccines should be administered to highly vaccinated populations in order to prevent progression of asymptomatic infected persons or mild COVID-19 caused by the Delta and Omicron variants to pneumonia. It will be necessary to conduct further follow-ups to determine the duration of protection against severe diseases after booster vaccinations.

## Figures and Tables

**Figure 1 vaccines-10-01215-f001:**
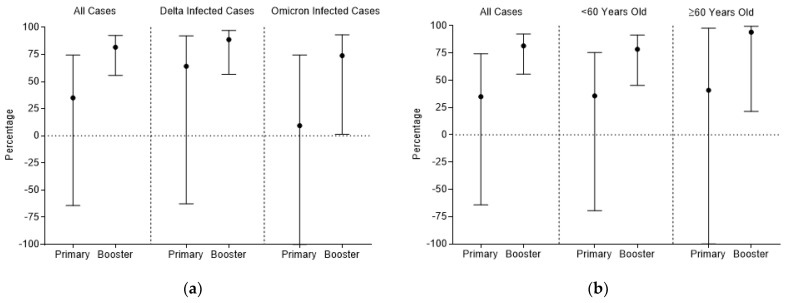
Protective effect of complete primary and booster of inactivated COVID-19 vaccines compared with the unvaccinated or partially vaccinated. (**a**) Protective effect against progression to pneumonia stratified by genotype; (**b**) protective effect against progression to pneumonia stratified by age.

**Table 1 vaccines-10-01215-t001:** Characteristics of the study population.

Characteristics	Clinical Outcome			Total	Statistics	*p*
	Asymptomatic	Mild	Moderate			
Genotype						
Delta	13 (59.1%)	36 (23.1%)	62 (71.3%)	111 (41.9%)	56.196	<0.001
Omicron	9 (40.9%)	120 (76.9%)	25 (28.7%)	154 (58.1%)		
Gender						
Male	14 (63.6%)	81 (51.9%)	52 (59.8%)	147 (55.5%)	2.04	0.361
Female	8 (36.4%)	75 (48.1%)	35 (40.2%)	118 (44.5%)		
Age						
Mean (95% CI)	38.9 (33.5–43.9)	36.1 (33.4–38.8)	49.3 (45.9–52.7)	40.7 (38.6–42.8)	18.103	<0.001
<60 years	22 (100%)	144 (92.3%)	71 (81.6%)	237 (89.4%)	9.599	0.008
≥60 years	(0%)	12 (7.7%)	16 (18.4%)	28 (10.6%)		
History of disease						
Hypertension						
None	19 (86.4%)	140 (89.7%)	68 (78.2%)	227 (85.7%)	6.11	0.047
Yes	3 (13.6%)	16 (10.3%)	19 (21.8%)	38 (14.3%)		
Diabetes						
None	22 (100%)	150 (96.2%)	73 (83.9%)	245 (92.5%)	13.962	0.001
Yes	(0%)	6 (3.8%)	14 (16.1%)	20 (7.5%)		
Heart disease						
None	22 (100%)	156 (100%)	85 (97.7%)	263 (99.2%)	4.123	0.127
Yes	(0%)	(0%)	2 (2.3%)	2 (0.8%)		
COPD						
None	22 (100%)	156 (100%)	87 (100%)	265 (100%)		
Asthma						
None	22 (100%)	155 (99.4%)	87 (100%)	264 (99.6%)	0.701	0.704
Yes	(0%)	1 (0.6%)	(0%)	1 (0.4%)		
Tumor						
None	22 (100%)	156 (100%)	86 (98.9%)	264 (99.6%)	2.054	0.358
Yes	(0%)	(0%)	1 (1.1%)	1 (0.4%)		
Autoimmune disease						
None	22 (100%)	154 (98.7%)	85 (97.7%)	261 (98.5%)	0.756	0.685
Yes	(0%)	2 (1.3%)	2 (2.3%)	4 (1.5%)		
Vaccination status						
Unvaccinated	2 (9.1%)	16 (10.3%)	20 (23%)	38 (14.3%)	16.137	0.013
Partially	(0%)	3 (1.9%)	4 (4.6%)	7 (2.6%)		
Complete primary	3 (13.6%)	46 (29.5%)	27 (31%)	76 (28.7%)		
Completely booster	17 (77.3%)	91 (58.3%)	36 (41.4%)	144 (54.3%)		
Duration of hospitalization						
Mean (95% CI)	15.5 (13.0–17.9)	15.3 (14.6–16.0)	17.9 (16.6–19.2)	16.2 (15.5–16.8)	6.673	0.001
Duration of Nucleic acid positive						
Mean (95% CI)	14.2 (11.3–17.1)	13.3 (12.4–14.0)	14.3 (13.1–15.6)	13.7 (13.0–14.3)	0.984	0.375
Receiving Oxygen therapy						
Yes	1 (4.5%)	1 (0.6%)	8 (9.2%)	10 (3.8%)	11.295	0.004
No	21 (95.5%)	155 (99.4%)	79 (90.8%)	255 (96.2%)		
Receiving intravenous drug use						
Yes	(0%)	5 (3.2%)	31 (35.6%)	36 (13.6%)	53.799	<0.001
No	22 (100%)	151 (96.8%)	56 (64.4%)	229 (86.4%)		
Total	22 (100%)	156 (100%)	87 (100%)	265 (100%)		

**Table 2 vaccines-10-01215-t002:** Association between pneumonia cases and vaccination stratified by genotype and age.

Characteristics	Non-Pneumonia Cases		Pneumonia Cases		OR (95% CI)	*p*
	(Asymptomatic + Mild)		(Moderate)			
	N	%	N	%		
All cases						
Vaccination status						
Unvaccinated or partially	21	11.8%	24	27.6%	Ref	
Complete primary	49	27.5%	27	31.0%	0.650 (0.257–1.643)	0.362
Completely booster	108	60.7%	36	41.4%	0.184 (0.076–0.444)	<0.001
Genotype						
Omicron	129	72.5%	25	28.7%		
Delta	49	27.5%	62	71.3%	13 (6.263–26.982)	<0.001
Age						
<60 years	166	93.3%	71	81.6%		
≥60 years	12	6.7%	16	18.4%	5.902 (1.852–18.809)	0.003
History of diabetes						
None	172	96.6%	73	83.9%		
Yes	6	3.4%	14	16.1%	4.888 (1.417–16.856)	0.012
History of hypertension						
None	159	89.3%	68	78.2%		
Yes	19	10.7%	19	21.8%	0.924 (0.377–2.263)	0.863
Delta infected cases						
Vaccination status						
Unvaccinated or partially	3	6.1%	17	27.4%	Ref	
Complete primary	8	16.3%	17	27.4%	0.359 (0.079–1.626)	0.184
Completely booster	38	77.6%	28	45.2%	0.113 (0.03–0.434)	0.001
Age						
<60 years	49	100.0%	56	90.3%		
≥60 years	0	0.0%	6	9.7%	No data	
History of diabetes						
None	47	95.9%	55	88.7%		
Yes	2	4.1%	7	11.3%	4.591 (0.849–24.832)	0.077
History of hypertension						
None	40	81.6%	50	80.6%		
Yes	9	18.4%	12	19.4%	0.787 (0.263–2.361)	0.67
Omicron infected cases						
Vaccination status						
Unvaccinated or partially	18	14.0%	7	28.0%	Ref	
Complete primary	41	31.8%	10	40.0%	0.906 (0.256–3.203)	0.878
Completely booster	70	54.3%	8	32.0%	0.261 (0.069–0.986)	0.048
Age						
<60 years	117	90.7%	15	60.0%		
≥60 years	12	9.3%	10	40.0%	4.004 (1.052–15.232)	0.042
History of diabetes						
None	125	96.9%	18	72.0%		
Yes	4	3.1%	7	28.0%	6.187 (1.198–31.943)	0.03
History of hypertension						
None	119	92.2%	18	72.0%		
Yes	10	7.8%	7	28.0%	1.413 (0.323–6.191)	0.646
Patients aged <60 years old						
Vaccination status						
Unvaccinated or partially	20	12.0%	17	23.9%	Ref	
Complete primary	48	28.9%	23	32.4%	0.643 (0.244–1.695)	0.372
Completely booster	98	59.0%	31	43.7%	0.215 (0.084–0.547)	0.001
Genotype						
Omicron	117	70.5%	15	21.1%		
Delta	49	29.5%	56	78.9%	11.578 (5.557–24.121)	<0.001
History of diabetes						
None	162	97.6%	66	93.0%		
Yes	4	2.4%	5	7.0%	3.733 (0.799–17.433)	0.094
History of hypertension						
None	153	92.2%	61	85.9%		
Yes	13	7.8%	10	14.1%	0.996 (0.367–2.706)	0.994
Patients aged ≥60 years old						
Vaccination status						
Unvaccinated or partially	1	8.3%	7	43.8%		
Complete primary	1	8.3%	4	25.0%	0.591 (0.022–15.863)	0.754
Completely booster	10	83.3%	5	31.3%	0.059 (0.004–0.785)	0.032
Genotype						
Omicron	12	100.0%	10	62.5%		
Delta	0	0.0%	6	37.5%	No data	
History of diabetes						
None	10	83.3%	7	43.8%		
Yes	2	16.7%	9	56.3%	8.861 (0.968–81.129)	0.053
History of hypertension						
None	6	50.0%	7	43.8%		
Yes	6	50.0%	9	56.3%	0.804 (0.099–6.522)	0.838

**Table 3 vaccines-10-01215-t003:** Association between vaccination and medical resources stratified by disease severity.

	Unvaccinated	Partially	Complete Primary	Completely Booster	*p*
Pneumonia group					
Duration of hospitalization					
Mean (95% CI)	21.7 (18.7, 24.7)	16.3 (9.3, 23.2)	17.7 (15.4, 20.1)	16.1 (14.3, 18.0)	0.009
Receiving Oxygen therapy					
Yes	5 (62.5%)	0 (0%)	3 (37.5%)	0 (0%)	0.012
No	15 (19%)	4 (5.1%)	24 (30.4%)	36 (45.6%)	
Receiving intravenous drug use					
Yes	0 (0%)	0 (0%)	0 (0%)	5 (100%)	0.224
No	18 (10.4%)	3 (1.7%)	49 (28.3%)	103 (59.5%)	
No-pneumonia group					
Duration of hospitalization					
Mean (95% CI)	14.6 (12.5, 16.7)	13.67 (9.9, 17.5)	14.80 (13.6, 16.0)	15.70 (14.8, 16.7)	0.533
Receiving Oxygen therapy					
Yes	1 (50%)	0 (0%)	1 (50%)	0 (0%)	0.099
No	17 (9.7%)	3 (1.7%)	48 (27.3%)	108 (61.4%)	
Receiving intravenous drug use					
Yes	10 (32.3%)	2 (6.5%)	10 (32.3%)	9 (29%)	0.428
No	10 (17.9%)	2 (3.6%)	17 (30.4%)	27 (48.2%)	

## Data Availability

The data that support the findings of this study are available from the corresponding author, [chenxy1998@hotmail.com (X.C.); wj81732@hotmail.com (J.W.)], upon reasonable request.

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
