# Peer review of "Protective Effect of Inactivated COVID-19 Vaccines against Progression of SARS-CoV-2 Omicron and Delta Variant Infections to Pneumonia in Beijing, China, in 2022"

_vaccines, 2022, doi:10.3390/vaccines10081215_

Round 1

Reviewer 1 Report

In their manuscript the authors provide evidence that 3 doses of an inactivated are necessary to reduce the risk of progression to pneumonia. Albeit of limited size, the study in general is of value and can be published after addressing the following comments:

Why were patients hospitalized if the majority of them had only mild/asymptomatic disease?

As the authors also use the length of the hospital stay as one parameter, it would be informative to know what are the discharge criteria as this could affect the results.

Please mention which vaccines (manufacturer) the patients received, i.e. not only type of vaccine.

line 43: administered or produced vaccine doses. Please clarify

line 58-60: The sentence reads awkward. Please rephrase.

Line 63: I don’t think that Omicron has an increased virulence. If anything studies show that the intrinsic virulence of Omicron is similar or lower than Delta.

Line 149: I assume the CI should be “1.643” instead of “-1.643”, otherwise it makes no sense.

Paragraph 3.4 (line165-171) doesn’t make a lot of sense to me. It’s a not really surprising that patients with pneumonia are staying longer in the hospital and receive oxygen more frequently. However, it would be interesting to see the effect of vaccination on medical resources. Could the authors please provide that.

Is time since last vaccine dose received similar between the group that received 2 doses and the group that received 3 doses? I would assume that time since last vaccination is shorter for the boosted group. Could the authors please discuss that the observed reduction in progression to pneumonia could also be a timing effect.

Could the authors please provide a clear definition what they consider as pneumonia? How was the pneumonia diagnosed?

Reviewer 2 Report

Manuscript by Li et al provides a snapshot at the Delta or Omicron variant COVID-19 progressing that leads to pneumonia in Beijing population. The study suggests that inactivated vaccines reduce the risk of pneumonia progression.

The study is straight forward observational, and not surprisingly the outcome is similarly to other vaccine studies. This study can be useful give the fact that it is represents the data from the one of the important Chinese cities. Although, an intense study and analysis is expected on is topic. There are several points a that this reviewer likes to bring to authors attention.

1.       There manuscript is mere observation and statistics, a modelling studies is expected to derive inferences especially with many variables (age, premedical conditions……..etc). All these associations needs to be tied-up to make a good inference (Table 1 and 2).  The Delta or Omicron variables need some attention too.

2.       The conclusions in the abstract, last para of introduction and “conclusion” section seems inconsistent. Please work on this.

3.       Some of the important data from the table2 can be represented as venn diagram, this makes it more useful to relate and associate.

4.       Figure 1 should be modified such that the individual data points are represented on the graph. Please show the individual points so that the range and density is visible.

Overall, this is short study that just showcase’s the observation with minor statistical analysis. There is a need for the conclusion can be reinforced with modelling this data. The data is under analyzed, there is much more that can be derived of this study. The intent is not to lose the message or create unnecessary confusion but to support the claimed conclusion.

Round 2

Reviewer 2 Report

Authors have improved the manuscript with the revision. Manuscript may be recommended to the Editor.